# Glow-TTS: A Generative Flow for Text-to-Speech via Monotonic Alignment Search

**Jaehyeon Kim**
Kakao Enterprise
jay.xyz@kakaoenterprise.com

**Sungwon Kim**
Data Science & AI Lab.
Seoul National University
ksw0306@snu.ac.kr

**Jungil Kong**
Kakao Enterprise
henry.k@kakaoenterprise.com

**Sungroh Yoon**∗
Data Science & AI Lab.
Seoul National University
sryoon@snu.ac.kr

## Abstract

Recently, text-to-speech (TTS) models such as FastSpeech and ParaNet have been proposed to generate mel-spectrograms from text in parallel. Despite the advantage, the parallel TTS models cannot be trained without guidance from autoregressive TTS models as their external aligners. In this work, we propose Glow-TTS, a flow-based generative model for parallel TTS that does not require any external aligner. By combining the properties of flows and dynamic programming, the proposed model searches for the most probable monotonic alignment between text and the latent representation of speech on its own. We demonstrate that enforcing hard monotonic alignments enables robust TTS, which generalizes to long utterances, and employing generative flows enables fast, diverse, and controllable speech synthesis. Glow-TTS obtains an order-of-magnitude speed-up over the autoregressive model, Tacotron 2, at synthesis with comparable speech quality. We further show that our model can be easily extended to a multi-speaker setting.

## 1 Introduction

Text-to-speech (TTS) is a task in which speech is generated from text, and deep-learning-based TTS models have succeeded in producing natural speech. Among neural TTS models, autoregressive models, such as Tacotron 2 [23] and Transformer TTS [13], have shown state-of-the-art performance. Despite the high synthesis quality of autoregressive TTS models, there are a few difficulties in deploying them directly in real-time services. As the inference time of the models grows linearly with the output length, undesirable delay caused by generating long utterances can be propagated to the multiple pipelines of TTS systems without designing sophisticated frameworks [14]. In addition, most of the autoregressive models show a lack of robustness in some cases [20, 16]. For example, when an input text includes repeated words, autoregressive TTS models sometimes produce serious attention errors.

To overcome such limitations of the autoregressive TTS models, parallel TTS models, such as FastSpeech [20], have been proposed. These models can synthesize mel-spectrograms significantly faster than the autoregressive models. In addition to the fast sampling, FastSpeech reduces the failure cases of synthesis, such as mispronouncing, skipping, or repeating words, by constraining its alignment to be monotonic. However, to train the parallel TTS models, well-aligned attention

---

∗Corresponding author

maps between text and speech are necessary. Recently proposed parallel models extract attention maps from their external aligners, pre-trained autoregressive TTS models [16, 20]. Therefore, the performance of the models critically depends on that of the external aligners.

In this work, we eliminate the necessity of any external aligner and simplify the training procedure of parallel TTS models. Here, we propose Glow-TTS, a flow-based generative model for parallel TTS that can internally learn its own alignment.

By combining the properties of flows and dynamic programming, Glow-TTS efficiently searches for the most probable monotonic alignment between text and the latent representation of speech. The proposed model is directly trained to maximize the log-likelihood of speech with the alignment. We demonstrate that enforcing hard monotonic alignments enables robust TTS, which generalizes to long utterances, and employing flows enables fast, diverse, and controllable speech synthesis.

Glow-TTS can generate mel-spectrograms 15.7 times faster than the autoregressive TTS model, Tacotron 2, while obtaining comparable performance. As for robustness, the proposed model outperforms Tacotron 2 significantly when input utterances are long. By altering the latent representation of speech, we can synthesize speech with various intonation patterns and regulate the pitch of speech. We further show that our model can be extended to a multi-speaker setting with only a few modifications. Our source code[2] and synthesized audio samples[3] are publicly available.

## 2 Related Work

**Alignment Estimation between Text and Speech.** Traditionally, hidden Markov models (HMMs) have been used to estimate unknown alignments between text and speech [19, 25]. In speech recognition, CTC has been proposed as a method of alleviating the downsides of HMMs, such as the assumption of conditional independence over observations, through a discriminative neural network model [6]. Both methods above can efficiently estimate alignments through forward-backward algorithms with dynamic programming. In this work, we introduce a similar dynamic programming method to search for the most probable alignment between text and the latent representation of speech, where our modeling differs from CTC in that it is generative, and from HMMs in that it can sample sequences in parallel without the assumption of conditional independence over observations.

**Text-to-Speech Models.** TTS models are a family of generative models that synthesize speech from text. TTS models, such as Tacotron 2 [23], Deep Voice 3 [17] and Transformer TTS [13], generate a mel-spectrogram from text, which is comparable to that of the human voice. Enhancing the expressiveness of TTS models has also been studied. Auxiliary embedding methods have been proposed to generate diverse speech by controlling factors such as intonation and rhythm [24, 32], and some works have aimed at synthesizing speech in the voices of various speakers [9, 5]. Recently, several works have proposed methods to generate mel-spectrogram frames in parallel. FastSpeech [20], and ParaNet [16] significantly speed up mel-spectrogram generation over autoregressive TTS models, while preserving the quality of synthesized speech. However, both parallel TTS models need to extract alignments from pre-trained autoregressive TTS models to alleviate the length mismatch problem between text and speech. Our Glow-TTS is a standalone parallel TTS model that internally learns to align text and speech by leveraging the properties of flows and dynamic programming.

**Flow-based Generative Models.** Flow-based generative models have received a lot of attention due to their advantages [7, 4, 21]. They can estimate the exact likelihood of the data by applying invertible transformations. Generative flows are simply trained to maximize the likelihood. In addition to efficient density estimation, the transformations proposed in [2, 3, 12] guarantee fast and efficient sampling. Prenger et al. [18] and Kim et al. [10] introduced these transformations for speech synthesis to overcome the slow sampling speed of an autoregressive vocoder, WaveNet [29]. Their proposed models both synthesized raw audio significantly faster than WaveNet. By applying these transformations, Glow-TTS can synthesize a mel-spectrogram given text in parallel.

In parallel with our work, AlignTTS [34], Flowtron [28], and Flow-TTS [15] have been proposed. AlignTTS and Flow-TTS are parallel TTS models without the need of external aligners, and Flowtron is a flow-based model which shows the ability of style transfer and controllability of speech variation. However, AlignTTS is not a flow-based model but a feed-forward network, and Flowtron and Flow-

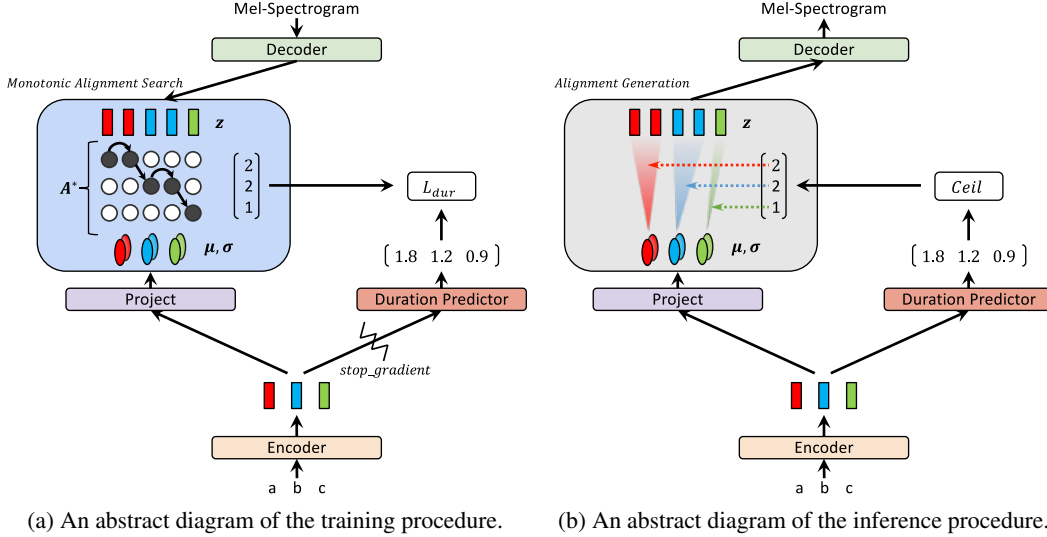

(a) An abstract diagram of the training procedure.　　(b) An abstract diagram of the inference procedure.

Figure 1: Training and inference procedures of Glow-TTS.

TTS use soft attention modules. By employing both hard monotonic alignments and generative flows, our model combines the best of both worlds in terms of robustness, diversity and controllability.

## 3  Glow-TTS

Inspired by the fact that a human reads out text in order, without skipping any words, we design Glow-TTS to generate a mel-spectrogram conditioned on a monotonic and non-skipping alignment between text and speech representations. In Section 3.1, we formulate the training and inference procedures of the proposed model, which are also illustrated in Figure 1. We present our alignment search algorithm in Section 3.2, which removes the necessity of external aligners from training, and the architecture of all components of Glow-TTS (i.e., the text encoder, duration predictor, and flow-based decoder) is covered in Section 3.3.

### 3.1  Training and Inference Procedures

Glow-TTS models the conditional distribution of mel-spectrograms $P_X(x|c)$ by transforming a conditional prior distribution $P_Z(z|c)$ through the flow-based decoder $f_{dec} : z \to x$, where $x$ and $c$ denote the input mel spectrogram and text sequence, respectively. By using the change of variables, we can calculate the exact log-likelihood of the data as follows:

$$\log P_X(x|c) = \log P_Z(z|c) + \log \left| \det \frac{\partial f_{dec}^{-1}(x)}{\partial x} \right| \tag{1}$$

We parameterize the data and prior distributions with network parameters $\theta$ and an alignment function $A$. The prior distribution $P_Z$ is the isotropic multivariate Gaussian distribution and all the statistics of the prior distribution, $\mu$ and $\sigma$, are obtained by the text encoder $f_{enc}$. The text encoder maps the text condition $c = c_{1:T_{text}}$ into the statistics, $\mu = \mu_{1:T_{text}}$ and $\sigma = \sigma_{1:T_{text}}$, where $T_{text}$ denotes the length of the text input. In our formulation, the alignment function $A$ stands for the mapping from the index of the latent representation of speech to that of statistics from $f_{enc}$: $A(j) = i$ if $z_j \sim N(z_j; \mu_i, \sigma_i)$. We assume the alignment function $A$ to be monotonic and surjective to ensure Glow-TTS not to skip or repeat the text input. Then, the prior distribution can be expressed as follows:

$$\log P_Z(z|c; \theta, A) = \sum_{j=1}^{T_{mel}} \log \mathcal{N}(z_j; \mu_{A(j)}, \sigma_{A(j)}), \tag{2}$$

where $T_{mel}$ denotes the length of the input mel-spectrogram.

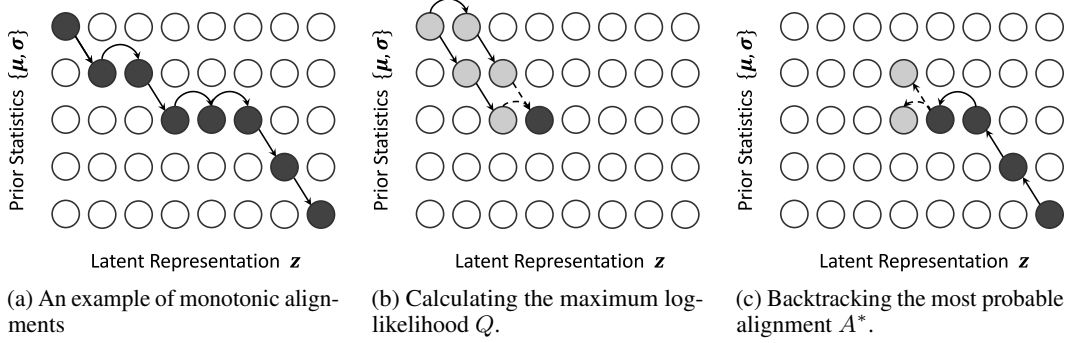

| (a) An example of monotonic align- | (b) Calculating the maximum log- | (c) Backtracking the most probable |
| ments | likelihood $Q$. | alignment $A^*$. |

Figure 2: Illustrations of the monotonic alignment search.

Our goal is to find the parameters $\theta$ and the alignment $A$ that maximize the log-likelihood of the data, as in Equation 3. However, it is computationally intractable to find the global solution. To tackle the intractability, we reduce the search space of the parameters and alignment by decomposing the objective into two subsequent problems: $(i)$ searching for the most probable monotonic alignment $A^*$ with respect to the current parameters $\theta$, as in Equation 4, and $(ii)$ updating the parameters $\theta$ to maximize the log-likelihood $\log p_X(x|c; \theta, A^*)$. In practice, we handle these two problems using an iterative approach. At each training step, we first find $A^*$, and then update $\theta$ using the gradient descent. The iterative procedure is actually one example of widely used Viterbi training [19], which maximizes log likelihood of the most likely hidden alignment. The modified objective does not guarantee the global solution of Equation 3, but it still provides a good lower bound of the global solution.

$$\max_{\theta, A} L(\theta, A) = \max_{\theta, A} \log P_X(x|c; A, \theta) \tag{3}$$

$$A^* = \arg\max_A \log P_X(x|c; A, \theta) = \arg\max_A \sum_{j=1}^{T_{mel}} \log \mathcal{N}(z_j; \mu_{A(j)}, \sigma_{A(j)}) \tag{4}$$

To solve the alignment search problem $(i)$, we introduce an alignment search algorithm, monotonic alignment search (MAS), which we describe in Section 3.2.

To estimate the most probable monotonic alignment $A^*$ at inference, we also train the duration predictor $f_{dur}$ to match the duration label calculated from the alignment $A^*$, as in Equation 5. Following the architecture of FastSpeech [20], we append the duration predictor on top of the text encoder and train it with the mean squared error loss (MSE) in the logarithmic domain. We also apply the stop gradient operator $sg[\cdot]$, which removes the gradient of input in the backward pass [30], to the input of the duration predictor to avoid affecting the maximum likelihood objective. The loss for the duration predictor is described in Equation 6.

$$d_i = \sum_{j=1}^{T_{mel}} 1_{A^*(j)=i}, i = 1, ..., T_{text} \tag{5}$$

$$L_{dur} = MSE(f_{dur}(sg[f_{enc}(c)]), d) \tag{6}$$

During inference, as shown in Figure 1b, the statistics of the prior distribution and alignment are predicted by the text encoder and duration predictor. Then, a latent variable is sampled from the prior distribution, and a mel-spectrogram is synthesized in parallel by transforming the latent variable through the flow-based decoder.

## 3.2 Monotonic Alignment Search

As mentioned in Section 3.1, MAS searches for the most probable monotonic alignment between the latent variable and the statistics of the prior distribution, which are came from the input speech and text, respectively. Figure 2a shows one example of possible monotonic alignments.

We present our alignment search algorithm in Algorithm 1. We first derive a recursive solution over partial alignments and then find the entire alignment.

Let $Q_{i,j}$ be the maximum log-likelihood where the statistics of the prior distribution and the latent variable are partially given up to the $i$-th and $j$-th elements, respectively. Then, $Q_{i,j}$ can be recursively formulated with $Q_{i-1,j-1}$ and $Q_{i,j-1}$, as in Equation 7, because if the last elements of partial sequences, $z_j$ and $\{\mu_i, \sigma_i\}$, are aligned, the previous latent variable $z_{j-1}$ should have been aligned to either $\{\mu_{i-1}, \sigma_{i-1}\}$ or $\{\mu_i, \sigma_i\}$ to satisfy monotonicity and surjection.

$$Q_{i,j} = \max_A \sum_{k=1}^{j} \log \mathcal{N}(z_k; \mu_{A(k)}, \sigma_{A(k)}) = \max(Q_{i-1,j-1}, Q_{i,j-1}) + \log \mathcal{N}(z_j; \mu_i, \sigma_i) \quad (7)$$

This process is illustrated in Figure 2b. We iteratively calculate all the values of $Q$ up to $Q_{T_{text}, T_{mel}}$.

Similarly, the most probable alignment $A^*$ can be obtained by determining which $Q$ value is greater in the recurrence relation, Equation 7. Thus, $A^*$ can be found efficiently with dynamic programming by caching all $Q$ values; all the values of $A^*$ are backtracked from the end of the alignment, $A^*(T_{mel}) = T_{text}$, as in Figure 2c.

The time complexity of the algorithm is $O(T_{text} \times T_{mel})$. Even though the algorithm is difficult to parallelize, it runs efficiently on CPU without the need for GPU executions. In our experiments, it spends less than 20 ms on each iteration, which amounts to less than 2% of the total training time. Furthermore, we do not need MAS during inference, as the duration predictor is used to estimate the alignment.

---

**Algorithm 1** Monotonic Alignment Search

---

**Input:** latent representation $z$, the statistics of prior distribution $\mu$, $\sigma$, the mel-spectrogram length $T_{mel}$, the text length $T_{text}$
**Output:** monotonic alignment $A^*$

Initialize $Q_{\cdot,\cdot} \leftarrow -\infty$, a cache to store the maximum log-likelihood calculations
Compute the first row $Q_{1,j} \leftarrow \sum_{k=1}^{j} \log \mathcal{N}(z_k; \mu_1, \sigma_1)$, for all $j$
**for** $j = 2$ **to** $T_{mel}$ **do**
    **for** $i = 2$ **to** $\min(j, T_{text})$ **do**
        $Q_{i,j} \leftarrow \max(Q_{i-1,j-1}, Q_{i,j-1}) + \log \mathcal{N}(z_j; \mu_i, \sigma_i)$
    **end for**
**end for**
Initialize $A^*(T_{mel}) \leftarrow T_{text}$
**for** $j = T_{mel} - 1$ **to** $1$ **do**
    $A^*(j) \leftarrow \arg\max_{i \in \{A^*(j+1)-1, A^*(j+1)\}} Q_{i,j}$
**end for**

---

### 3.3 Model Architecture

Each component of Glow-TTS is briefly explained in this section, and the overall model architecture and model configurations are shown in Appendix A.

**Decoder.** The core part of Glow-TTS is the flow-based decoder. During training, we need to efficiently transform a mel-spectrogram into the latent representation for maximum likelihood estimation and our internal alignment search. During inference, it is necessary to transform the prior distribution into the mel-spectrogram distribution efficiently for parallel decoding. Therefore, our decoder is composed of a family of flows that can perform forward and inverse transformation in parallel. Specifically, our decoder is a stack of multiple blocks, each of which consists of an activation normalization layer, invertible 1x1 convolution layer, and affine coupling layer. We follow the affine coupling layer architecture of WaveGlow [18], except we do not use the local conditioning [29].

For computational efficiency, we split 80-channel mel-spectrogram frames into two halves along the time dimension and group them into one 160-channel feature map before the flow operations. We also modify 1x1 convolution to reduce the time-consuming calculation of the Jacobian determinant.

Before every 1x1 convolution, we split the feature map into 40 groups along the channel dimension and perform 1x1 convolution on them separately. To allow channel mixing in each group, the same number of channels are extracted from one half of the feature map separated by coupling layers and the other half, respectively. A detailed description can be found in Appendix A.1.

**Encoder and Duration Predictor.** We follow the encoder structure of Transformer TTS [13] with two slight modifications. We remove the positional encoding and add relative position representations [22] into the self-attention modules instead. We also add a residual connection to the encoder pre-net. To estimate the statistics of the prior distribution, we append a linear projection layer at the end of the encoder. The duration predictor is composed of two convolutional layers with ReLU activation, layer normalization, and dropout followed by a projection layer. The architecture and configuration of the duration predictor are the same as those of FastSpeech [20].

## 4 Experiments

To evaluate the proposed methods, we conduct experiments on two different datasets. For the single speaker setting, a single female speaker dataset, LJSpeech [8], is used, which consists of 13,100 short audio clips with a total duration of approximately 24 hours. We randomly split the dataset into the training set (12,500 samples), validation set (100 samples), and test set (500 samples). For the multi-speaker setting, the train-clean-100 subset of the LibriTTS corpus [33] is used, which consists of audio recordings of 247 speakers with a total duration of about 54 hours. We first trim the beginning and ending silence of all the audio clips and filter out the data with text lengths over 190. We then split it into the training (29,181 samples), validation (88 samples), and test sets (442 samples). Additionally, out-of-distribution text data are collected for the robustness test. Similar to [1], we extract 227 utterances from the first two chapters of the book *Harry Potter and the Philosopher's Stone*. The maximum length of the collected data exceeds 800.

We compare Glow-TTS with the best publicly available autoregressive TTS model, Tacotron 2 [26]. For all the experiments, phonemes are chosen as input text tokens. We follow the configuration for the mel-spectrogram of [27], and all the generated mel-spectrograms from both models are transformed to raw waveforms through the pre-trained vocoder, WaveGlow [27].

During training, we simply set the standard deviation $\sigma$ of the learnable prior to be a constant 1. Glow-TTS was trained for 240K iterations using the Adam optimizer [11] with the Noam learning rate schedule [31]. This required only 3 days with mixed precision training on two NVIDIA V100 GPUs.

To train muli-speaker Glow-TTS, we add the speaker embedding and increase the hidden dimension. The speaker embedding is applied in all affine coupling layers of the decoder as a global conditioning [29]. The rest of the settings are the same as for the single speaker setting. For comparison, We also trained Tacotron 2 as a baseline, which concatenates the speaker embedding with the encoder output at each time step. We use the same model configuration as the single speaker one. All multi-speaker models were trained for 960K iterations on four NVIDIA V100 GPUs.

## 5 Results

### 5.1 Audio Quality

We measure the mean opinion score (MOS) via Amazon Mechanical Turk to compare the quality of all audio clips, including ground truth (GT), and the synthesized samples; 50 sentences are randomly chosen from the test set for the evaluation. The results are shown in Table 1. The quality of speech converted from the GT mel-spectrograms by the vocoder (4.19±0.07) is the upper limit of the TTS models. We vary the standard deviation (i.e., temperature $T$) of

Table 1: The Mean Opinion Score (MOS) of single speaker TTS models with 95% confidence intervals.

| Method | 9-scale MOS |
|---|---|
| GT | $4.54 \pm 0.06$ |
| GT (Mel + WaveGlow) | $4.19 \pm 0.07$ |
| Tacotron2 (Mel + WaveGlow) | $3.88 \pm 0.08$ |
| Glow-TTS ($T = 0.333$, Mel + WaveGlow) | $4.01 \pm 0.08$ |
| Glow-TTS ($T = 0.500$, Mel + WaveGlow) | $3.96 \pm 0.08$ |
| Glow-TTS ($T = 0.667$, Mel + WaveGlow) | $3.97 \pm 0.08$ |

the prior distribution at inference; Glow-TTS shows the best performance at the temperature of 0.333.

For any temperature, it shows comparable performance to Tacotron 2. We also analyze side-by-side evaluation between Glow-TTS and Tacotron 2. The result is shown in Appendix B.2.

## 5.2 Sampling Speed and Robustness

**Sampling Speed.** We use the test set to measure the sampling speed of the models. Figure 3a demonstrates that the inference time of our model is almost constant at 40ms, regardless of the length, whereas that of Tacotron 2 linearly increases with the length due to the sequential sampling. On average, Glow-TTS shows a 15.7 times faster synthesis speed than Tacotron 2.

We also measure the total inference time for synthesizing 1-minute speech in an end-to-end setting with Glow-TTS and WaveGlow. The total inference time to synthesize the 1-minute speech is only 1.5 seconds[4] and the inference time of Glow-TTS and WaveGlow accounts for 4% and 96% of the total inference time, respectively; the inference time of Glow-TTS takes only 55ms to synthesize the mel-spectrogram, which is negligible compared to that of the vocoder.

**Robustness.** We measure the character error rate (CER) of the synthesized samples from long utterances in the book *Harry Potter and the Philosopher's Stone* via the Google Cloud Speech-To-Text API.[5] Figure 3b shows that the CER of Tacotron 2 starts to grow when the length of input characters exceeds about 260. On the other hand, even though our model has not seen such long texts during training, it shows robustness to long texts. We also analyze attention errors on specific sentences. The results are shown in Appendix B.1.

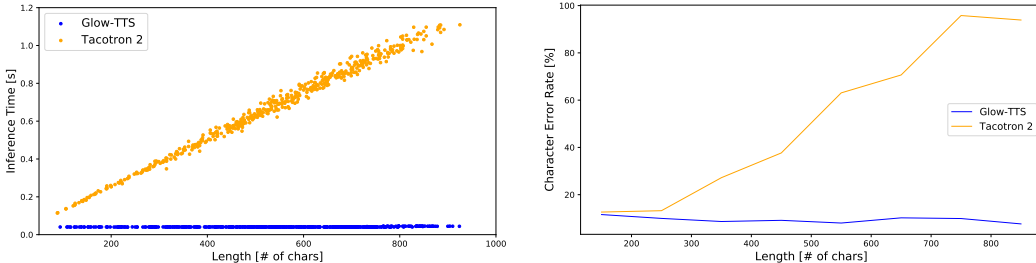

(a) The inference time comparison for Tacotron 2 and Glow-TTS (yellow: Tacotron2, blue: Glow-TTS).

(b) Robustness to the length of input utterances (yellow: Tacotron2, blue: Glow-TTS).

Figure 3: Comparison of inference time and length robustness.

## 5.3 Diversity and Controllability

Because Glow-TTS is a flow-based generative model, it can synthesize diverse samples; each latent representation $z$ sampled from an input text is converted to a different mel-spectrogram $f_{dec}(z)$. Specifically, the latent representation $z \sim \mathcal{N}(\mu, T)$ can be expressed as follows:

$$z = \mu + \epsilon * T \tag{8}$$

where $\epsilon$ denotes a sample from the standard normal distribution and $\mu$ and $T$ denote the mean and standard deviation (i.e., temperature) of the prior distribution, respectively.

To decompose the effect of $\epsilon$ and $T$, we draw pitch tracks of synthesized samples in Figure 4 by varying $\epsilon$ and $T$ one at a time. Figure 4a demonstrates that diverse stress or intonation patterns of speech arise from $\epsilon$, whereas Figure 4b demonstrates that we can control the pitch of speech while maintaining similar intonation by only varying $T$. Additionally, we can control speaking rates of speech by multiplying a positive scalar value across the predicted duration of the duration predictor. The result is visualized in Figure 5; the values multiplied by the predicted duration are 1.25, 1.0, 0.75, and 0.5.

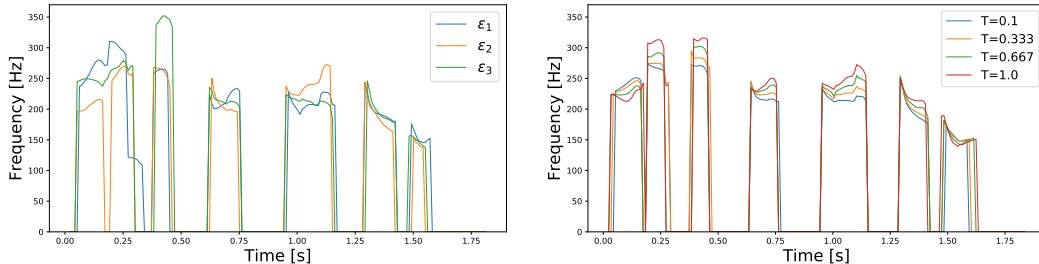

(a) Pitch tracks of the generated speech samples from the same sentence with different gaussian noise $\epsilon$ and the same temperature $T = 0.667$.

(b) Pitch tracks of the generated speech samples from the same sentence with different temperatures $T$ and the same gaussian noise $\epsilon$.

Figure 4: The fundamental frequency (F0) contours of synthesized speech samples from Glow-TTS trained on the LJ dataset.

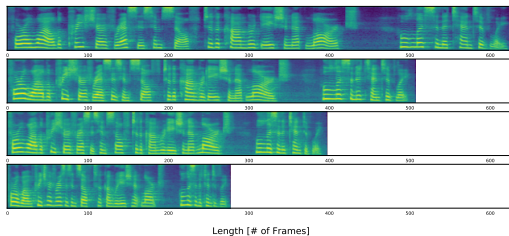

Figure 5: Mel-spectrograms of the generated speech samples with different speaking rates.

## 5.4 Multi-Speaker TTS

**Audio Quality.** We measure the MOS as done in Section 5.1; we select 50 speakers, and randomly sample one utterance per a speaker from the test set for evaluation. The results are presented in Table 2. The quality of speech converted from the GT mel-spectrograms (4.22±0.07) is the upper limit of the TTS models. Our model with the best configuration achieves 3.45 MOS, which results in comparable performance to Tacotron 2.

Table 2: The Mean Opinion Score (MOS) of a multi-speaker TTS with 95% confidence intervals.

| Method | 9-scale MOS |
| --- | --- |
| GT | $4.54 \pm 0.07$ |
| GT (Mel + WaveGlow) | $4.22 \pm 0.07$ |
| Tacotron2 (Mel + WaveGlow) | $3.35 \pm 0.12$ |
| Glow-TTS ($T = 0.333$, Mel + WaveGlow) | $3.20 \pm 0.12$ |
| Glow-TTS ($T = 0.500$, Mel + WaveGlow) | $3.31 \pm 0.12$ |
| Glow-TTS ($T = 0.667$, Mel + WaveGlow) | $3.45 \pm 0.11$ |

**Speaker-Dependent Duration.** Figure 6a shows the pitch tracks of generated speech from the same sentence with different speaker identities. As the only difference in input is speaker identities, the result demonstrates that our model differently predicts the duration of each input token with respect to the speaker identities.

**Voice Conversion.** As we do not provide any speaker identity into the encoder, the prior distribution is forced to be independent from speaker identities. In other words, Glow-TTS learns to disentangle the latent representation $z$ and the speaker identities. To investigate the degree of disentanglement, we transform a GT mel-spectrogram into the latent representation with the correct speaker identity and then invert it with different speaker identities. The detailed method can be found in Appendix B.3 The results are presented in Figure 6b. It shows that converted samples have different pitch levels while maintaining a similar trend.

## 6 Conclusion

In this paper, we proposed a new type of parallel TTS model, Glow-TTS. Glow-TTS is a flow-based generative model that is directly trained with maximum likelihood estimation. As the proposed model finds the most probable monotonic alignment between text and the latent representation of speech

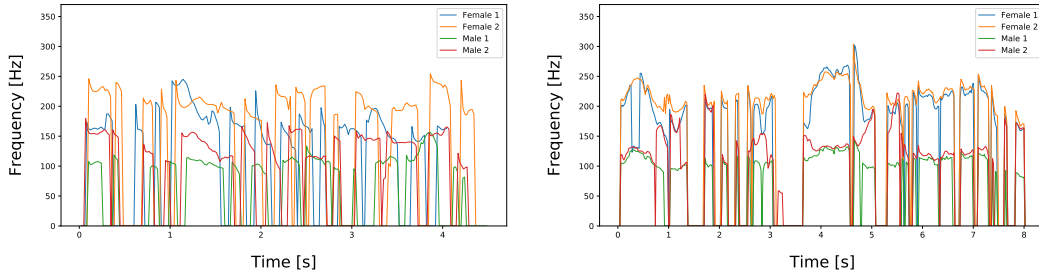

(a) Pitch tracks of the generated speech samples from the same sentence with different speaker identities.

(b) Pitch tracks of the voice conversion samples with different speaker identities.

Figure 6: The fundamental frequency (F0) contours of synthesized speech samples from Glow-TTS trained on the LibriTTS dataset.

on its own, the entire training procedure is simplified without the necessity of external aligners. In addition to the simple training procedure, we showed that Glow-TTS synthesizes mel-spectrograms 15.7 times faster than the autoregressive baseline, Tacotron 2, while showing comparable performance. We also demonstrated additional advantages of Glow-TTS, such as the ability to control the speaking rate or pitch of synthesized speech, robustness, and extensibility to a multi-speaker setting. Thanks to these advantages, we believe the proposed model can be applied in various TTS tasks such as prosody transfer or style modeling.

## Broader Impact

In this paper, researchers introduce Glow-TTS, a diverse, robust and fast text-to-speech (TTS) synthesis model. Neural TTS models including Glow-TTS, could be applied in many applications which require naturally synthesized speech. Some of the applications are AI voice assistant services, audiobook services, advertisements, automotive navigation systems and automated answering services. Therefore, by utilizing the models for synthesizing natural sounding speech, the providers of such applications could improve user satisfaction. In addition, the fast synthesis speed of the proposed model could be beneficial for some service providers who provide real time speech synthesis services. However, because of the ability to synthesize natural speech, the TTS models could also be abused through cyber crimes such as fake news or phishing. It means that TTS models could be used to impersonate voices of celebrities for manipulating behaviours of people, or to imitate voices of someone's friends or family for fraudulent purposes. With the development of speech synthesis technology, the growth of studies to detect real human voice from synthesized voices seems to be needed. Neural TTS models could sometimes synthesize undesirable speech with slurry or wrong pronunciations. Therefore, it should be used carefully in some domain where even a single pronunciation mistake is critical such as news broadcast. Additional concern is about the training data. Many corpus for speech synthesis contain speech data uttered by a handful of speakers. Without the detailed consideration and restriction about the range of uses the TTS models have, the voices of the speakers could be overused than they might expect.

## Acknowledgments

We would like to thank Jonghoon Mo, Hyeongseok Oh, Hyunsoo Lee, Yongjin Cho, Minbeom Cho, Younghun Oh, and Jaekyoung Bae for helpful discussions and advice. This work was partially supported by the BK21 FOUR program of the Education and Research Program for Future ICT Pioneers, Seoul National University in 2020.

## Footnotes

[2]https://github.com/jaywalnut310/glow-tts.

[3]https://jaywalnut310.github.io/glow-tts-demo/index.html.

[4]We generated a speech sample from our abstract paragraph, which we mention as the 1-minute speech.

[5]https://cloud.google.com/speech-to-text

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
