[Supplementary Material]

# Supplementary Material of
# Glow-TTS: A Generative Flow for Text-to-Speech via Monotonic Alignment Search

## Appendix A

### A.1. Details of the Model Architecture

The detailed encoder architecture is depicted in Figure 7. Some implementation details that we use in the decoder, and the decoder architecture are depicted in Figure 8.

We design the grouped 1x1 convolutions to be able to mix channels. For each group, the same number of channels are extracted from one half of the feature map separated by coupling layers and the other half, respectively. Figure 8c shows an example. If a coupling layer divides a 8-channel feature map [a, b, g, h, m, n, s, t] into two halves [a, b, g, h] and [m, n, s, t], we implement to group them into [a, b, m, n] and [g, h, s, t] when the number of groups is 2, or [a, m], [b, n], [g, s], [h, t] when the number of groups is 4.

Figure 7: The encoder architecture of Glow-TTS. The encoder gets a text sequence and processes it through the encoder pre-net and Transformer encoder. Then, the last projection layer and duration predictor of the encoder use the hidden representation $h$ to predict the statistics of prior distribution and duration, respectively.

(a) The decoder architecture of Glow-TTS. The decoder gets a mel-spectrogram and squeezes it. The, the decoder processes it through a number of flow blocks. Each flow block contains activation normalization layer, affine coupling layer, and invertible 1x1 convolution layer. The decoder reshapes the output to make equal to the input size.

(b) An illustration of $Squeeze$ and $UnSqueeze$ operations. When squeezing, the channel size doubles up and the number of time steps becomes a half. If the number of time steps is odd, we simply ignore the last element of mel-spectrogram sequence. It corresponds to about 11 ms audio, which makes no difference in quality.

(c) An illustration of our invertible 1x1 convolution. Two partitions used for coupling layers are colored blue and white, respectively. If input channel size is 8 and the number of groups is 2, we share a small 4x4 matrix as a kernel of the invertible 1x1 convolution layer. After channel mixing, we split the input into each group, and perform 1x1 convolution separately.

Figure 8: The decoder architecture of Glow-TTS and the implementation details used in the decoder.

## A.2. Hyper-parameters

Hyper-parameters of Glow-TTS are listed in Table 3. Contrary to the prevailing thought that flow-based generative models need the huge number of parameters, the total number of parameters of Glow-TTS (28.6M) is lower than that of FastSpeech (30.1M).

Table 3: Hyper-parameters of Glow-TTS.

| Hyper-Parameter | Glow-TTS (LJ Dataset) |
|---|---|
| Embedding Dimension | 192 |
| Pre-net Layers | 3 |
| Pre-net Hidden Dimension | 192 |
| Pre-net Kernel Size | 5 |
| Pre-net Dropout | 0.5 |
| Encoder Blocks | 6 |
| Encoder Multi-Head Attention Hidden Dimension | 192 |
| Encoder Multi-Head Attention Heads | 2 |
| Encoder Multi-Head Attention Maximum Relative Position | 4 |
| Encoder Conv Kernel Size | 3 |
| Encoder Conv Filter Size | 768 |
| Encoder Dropout | 0.1 |
| Duration Predictor Kernel Size | 3 |
| Duration Predictor Filter Size | 256 |
| Decoder Blocks | 12 |
| Decoder Activation Norm Data-dependent Initialization | True |
| Decoder Invertible 1x1 Conv Groups | 40 |
| Decoder Affine Coupling Dilation | 1 |
| Decoder Affine Coupling Layers | 4 |
| Decoder Affine Coupling Kernel Size | 5 |
| Decoder Affine Coupling Filter Size | 192 |
| Decoder Dropout | 0.05 |
| Total Number of Parameters | 28.6M |

# Appendix B

## B.1. Attention Error Analysis

Table 4: Attention error counts for TTS models on the 100 test sentences.

| Model | Attention Mask | Repeat | Mispronounce | Skip | Total |
|---|---|---|---|---|---|
| DeepVoice 3 [16] | X | 12 | 10 | 15 | 37 |
| DeepVoice 3 [16] | O | 1 | 4 | 3 | 8 |
| ParaNet [16] | X | 1 | 4 | 7 | 12 |
| ParaNet [16] | O | 2 | 4 | 0 | 6 |
| Tacotron 2 | X | 0 | 2 | 1 | 3 |
| Glow-TTS ($T = 0.333$) | X | 0 | 3 | 1 | 4 |

We measured attention alignment results using 100 test sentences used in ParaNet [16]. The average length and maximum length of test sentences are 59.65 and 315, respectively. Results are shown in Table 4. The results of DeepVoice 3 and ParaNet are taken from [16] and are not directly comparable due to the difference of grapheme-to-phoneme conversion tools.

Attention mask [16] is a method of computing attention only over a fixed window around target position at inference time. When constraining attention to be monotonic by applying attention mask technique, models make fewer attention errors.

Tacotron 2, which uses location sensitive attention, also makes little attention errors. Though Glow-TTS perform slightly worse than Tacotron 2 on the test sentences, Glow-TTS does not lose its robustness to extremely long sentences while Tacotron 2 does as we show in Section 5.2.

## B.2. Side-by-side Evaluation between Glow-TTS and Tacotron 2

We conducted 7-point CMOS evaluation between Tacotron 2 and Glow-TTS with the sampling temperature 0.333, which are both trained on the LJSpeech dataset. Through 500 ratings on 50 items, Glow-TTS wins Tacotron 2 by a gap of 0.934 as in Table 5, which shows preference towards our model over Tacotron 2.

Table 5: The Comparative Mean Opinion Score (CMOS) of single speaker TTS models

| Model | CMOS |
|---|---|
| Tacotron 2 | 0 |
| Glow-TTS ($T = 0.333$) | 0.934 |

## B.3. Voice Conversion Method

To transform a mel-spectrogram $x$ of a source speaker $s$ to a target mel-spectrogram $\hat{x}$ of a target speaker $\hat{s}$, we first find the latent representation $z$ through the inverse pass of the flow-based decoder $f_{dec}$ with the source speaker identity $s$.

$$z = f_{dec}^{-1}(x|s) \tag{9}$$

Then, we get the target mel-spectrogram $\hat{x}$ through the forward pass of the decoder with the target speaker identity $\hat{s}$.

$$\hat{x} = f_{dec}(z|\hat{s}) \tag{10}$$