[Reviews · NeurIPS 2020]

Review 1

Summary and Contributions: Glow-TTS is a non-autogressive flow-based TTS system that uses a 1-best Viterbi alignment between the text and output frames during training. Because training does not marginalize over alignments, the objective is only an approximation of the log-likelihood of the data. During training, the bulk of the computation can be done in parallel, with only the Viterbi alignment search requiring quadratic time. During inference, phoneme durations are predicted using using a feedforward net on top of the Transformer-based text encoder. The duration predictions are trained using an MSE objective. Synthesis can be done in a fully parallel manner due to the use of a Transformer-based encoder, non-autoregressive duration prediction, and non-autoregressive flows in the decoder. Unlike previous approaches to parallel TTS, Glow-TTS stands out in that it doesn't require an autoregressive teacher model, which simplifies the training procedure (The authors do point out the existence of parallel work from other groups that fit this description as well.)

Strengths: As mentioned above, the alignment mechanism is novel in the realm of end-to-end TTS; however it does bear some resemblance to traditional HMM-based TTS alignment schemes. Glow-TTS instead uses uniform transition probabilities and a 1-best alignment rather than marginalizing over possible alignments. This simplification turns out to work with their system empirically, though it means the objective isn't the exact log-likelihood (which is one of the advantages of using flow-based systems). The channel grouping scheme in the 1x1 convolutions seems novel and likely leads to significant efficiency and stability improvements; however, I do have some questions about how it interacts with the coupling layers and how it might affect the overall power of the model. Overall, the results are fairly strong. Publicly available datasets are used, and standard TTS evaluation methodologies are employed. MOS results are strong for a parallel system, and the sample temperature sweep is appreciated. Sampling speed is reported (a crucial justification for the quality hit when using a parallel approach) and shows constant synthesis time for any length of utterance. Additionally, robustness measure used in previous work shows that intelligibility is maintained for very long utterances. There is also a visualization of the pitch track variability achieved when sampling multiple realization of the same utterance.

Weaknesses: I was a little confused about how the grouped 1x1 convolutions interact with the coupling layers. If the standard (half-and-half) partitioning is used for the coupling layers and the grouped 1x1 convolutions never mix channels outside of their group of 4, then half of the channels will never be transformed by any coupling layer. I'm assuming the authors deal with this issue somehow (since the results are good), but I only briefly scanned the code and didn't want to work through all of the index gymnastics. I could see readers being confused by these missing details. Update: In their response, the authors said they will explain more of the details of the grouped 1x1 convolutions in their revised version. Listening to the audio examples, the quality is noticeably worse compared to state-of-the-art autoregressive models (which is expected), and it's most noticeable on longer utterances where the naturalness deteriorates significantly. Non-autoregressive duration prediction (which is trained using MSE) surely plays a role here due to the averaging effect of the MSE objective. Also, the finite size of the receptive fields in the flow could play a role here as well. While the results show visualization of multiple realization of a single utterance, it also points to the clear downside that the deterministic duration predictions disallow any sort of timing variations from sample-to-sample. Since both rhythm and pitch are important aspects of prosody, the model's ability to sample multiple realizations will be of limited usefulness. To address this, it could be interesting to apply a non-autoregressive flow to build a joint model of durations that could be sampled from at test time. Many of the audio samples seem to have a subtle whistling noise in the background, but I'm guessing this can be attributed to the WaveGlow vocoder as I've heard these artifacts in other work that uses the same vocoder. While parallel TTS is definitely an interesting problem, I question its importance (at least at the present moment). The primary computational bottleneck in mel spectrogram-based end-to-end TTS systems at present seems to be the neural vocoder (though parallel approaches to *that* problem might soon change that). Still, the ability to generate very long utterances in constant time (if they can fit in memory) is definitely desirable for long-form applications like book or news reading. Update: The authors make a good point that WaveGlow and other contemporary parallel neural vocoders have gotten to the point where mel spectrogram generation is becoming a significant portion of the computation. Autoregressive vocoders like WaveNet and WaveRNN still produce superior quality in my opinion, but the gap will likely narrow in the near future.

Correctness: With the exception of my doubts about how the grouped 1x1 convolutions interact with the coupling layers, the rest of the material seems to be correct to the best of my knowledge.

Clarity: This paper is fairly easy to understand and lays its information out in a logical way.

Relation to Prior Work: The authors mention that there is some contemporaneous work that has some similarities to this work. These are briefly mentioned, but it might be worthwhile to add a little more about AlignTTS since it seems to be the most similar. Past work is adequately covered.

Reproducibility: Yes

Additional Feedback: Minor edits: * Make sure to fill in the non-arXiv publication venues where appropriate in your references. I know Google Scholar tends to favor the arXiv reference for most paper, but it's useful to the reader to see the full attributions. * I noticed a few minor grammar errors. Please have your submission proof-read for English style and grammar issues.


Review 2

Summary and Contributions: This paper proposes speech synthesis using a generative flow model: a variational autoencoder with a bijective audio embedding, in which the bijection is used to convert data likelihoods into embedding likelihoods. Other than the generative flow idea, other key concepts are very similar to the methods used for HMM-based speech synthesis, back before tacotron and other recurrent deep models. The duration of each phone is computed by a separate RNN over the input embeddings. During training, the alignment between frames and phones is computed using a dynamic programming algorithm which seems to be identical to that of a hybrid HMM-DNN, though the authors don't seem to be aware of the similarity. Indeed, the whole system is exactly equal to a VAE-HMM (Ebbers et al., Interspeech 2017), except that the bijection allows the authors to turn the system from a speech recognizer into a speech synthesizer.

Strengths: The key innovation, really, is that bijection (the generative flow): that allows the authors to use the best ideas from both HMM-based and RNN-based speech synthesis.

Weaknesses: The theoretical weakness is that the authors don't compare their methods to HMM-based speech synthesis, hybrid HMM-DNN speech recognizers, or VAE-HMM. This doesn't really detract from the innovative idea, but it detracts from the experimental results, because experimental results are not really compared to either theoretically comparable or state of the art speech synthesizers.

Correctness: Experimental results beating a reasonable baseline (Tacotron-2) are presented for the single-speaker case, but experimental results for the multi-speaker case only compare different ablated and oracle versions of the proposed algorithm. Since multi-speaker TTS is pretty well established now (e.g., Deep Voice 2, NIPS 2019), it seems unreasonable to present multi-speaker results without a SOTA baseline.

Clarity: The paper is very clearly written.

Relation to Prior Work: The authors seem aware of a few of the most important TTS papers from, say, 2016-2018, but have missed large sections of the TTS literature that are relevant to their work either because they are theoretically related (the work on VAE-HMM and the work on HMM-based speech synthesis) or because they are the current state of the art (multi-speaker RNN-based and transformer-based TTS).

Reproducibility: Yes

Additional Feedback: None


Review 3

Summary and Contributions: The authors present a novel application of normalizing flows to the generation of mel spectrograms conditioned on sequences of text named Glow-TTS. In contrast to autoregressive text-to-spectrogram generators such as Tacotron 2, Glow TTS is able to synthesize spectrograms in parallel resulting in reported sampling speedups of 15x. In contrast to other parallel text-to-spectrogram generators such as FastSpeech and ParaNet, Glow-TTS does not depend on external aligners such as autoregressive teacher models or HMM alignment systems. The authors demonstrate robustness to out of distribution text, especially very long utterances. They demonstrate that Glow-TTS works well in a multi-speaker setting, and that it exhibits some degree of controllability (duration and pitch). The authors demonstrate Glow-TTS can be used as a voice conversion tool.

Strengths: The work alleviates issues with previously proposed parallel text-to-spectrogram models such as FastSpeech and ParaNet: * Glow-TTS does not require an external aligner or teacher model. * The loss function does not make erroneous independence assumptions across time and frequency as FastSpeech does. This work improves over autoregressive models such as Tacotron 2: * Sampling speed does not scale as poorly with sequence length as the spectrogram can be produced in parallel (limited only by the parallelism/capacity of the CPU or GPU generating the sequence). * Glow-TTS is robust to longer utterances than the Tacotron 2 baseline. The authors demonstrate how to extend Glow-TTS to a multi-speaker setting, and also demonstrate the ability to leverage the invertibility of the normalizing flow to achieve voice conversion. As such the work is a novel advancement over previous techniques, and is of interest to researchers and practitioners of normalizing flows and text-to-speech.

Weaknesses: * Monotonic Alignment Search algorithm used in training is deterministic and the training procedure doesn't represent uncertainty over possible alignments. At sampling time, the duration prediction module is also deterministic. Since it was trained with mean-squared-error, it is unable to produce natural and varied prosody since it is equivalent to taking the mean of a univariate Gaussian modeling the duration of each input token. Figure 4 is illustrative of this lack of diversity. The durations across multiple samples from the model are identical. * Section 3.1: "The modified objective does not guarantee the global solution of Equation 3, but it still provides a good lower bound of the global solution." How do we know this is a good lower bound? Please explicate. * Controllability in TTS is only of interest to TTS practitioners if the dimension of control offered is useful for downstream tasks (e.g. prosody/style transfer from a reference, control of emotion/affect/valence/arousal, achieving a desired meaning/prosody e.g. skepticism/confidence/uncertainty/...). While it is interesting that the sampling temperature of the latent prior seems to directly control pitch, I do not think that control via such an uncalibrated parameter is useful for a downstream task without a lot of work. The role pitch plays in speech is highly complicated and varies across language -- in some languages, it corresponds to lexical meaning, in others it's more prosodic. I do not think it's appropriate to describe Glow-TTS as controllable in its current form and would require discussion / comparison to other works that address this topic. Duration control is relevant and useful for downstream tasks. However, the method of control described in the manuscript is not novel (it has been used for over a decade in parametric TTS before end-to-end TTS became popular) and would be straightforward to implement with any TTS system with explicit durations. * I am unsure of the authors' claim that Glow-TTS produces diverse samples. Especially in light of the deterministic duration predictor and MAS algorithm, I would be surprised to see a model like this produce diverse prosody. Figure 4 is particularly revealing, since the pitch and durations across multiple samples from the model look and sound largely identical. To help support this claim, please include multiple samples from the model at various temperatures for a variety of text samples. Ideally the text would have some ambiguity over the intended meaning and the samples would be noticeably different from each other while remaining natural. * The multi-speaker results sound somewhat unnatural and the lower MOS confirms it. It sound almost as if the duration prediction module is underfit. The speaker dependent variation section presents a plot of variation across speakers but the audio samples undermine this. Do you know why this is? Please include a Tacotron 2 baseline in Table 2.

Correctness: * Please do not use MOS to compare systems especially when they are highly similar to each other. Please us a 7-point AB test (sometimes called "comparative MOS"). See Section 2.1: https://ecs.utdallas.edu/loizou/cimplants/quality_assessment_chapter.pdf * Please describe the methodology for running multi-speaker MOS evaluations. Did you sample the test sentences at random (over-representing more dominant speakers) or did you balance the number of samples from each speaker? Which speakers were evaluated? * To make the work more reproducible, please include the test phrases used for all MOS evaluation in the supplemental material. * Please include the text of the utterance in Figure 4, 5 and 6. * The performance optimization in the permutation layer seems flawed on its face, because the typical implementation of a RealNVP coupling layer leaves one half of the channels dimension untransformed and each block alternates the side it transforms. In Glow, the introduction of the permutation layer removed the need to alternate which half is transformed, leaving the permutation process to the network to learn. By factoring the channels into groups as the paper describes, no channel mixing from one half of the channels to the other can occur. However, upon reading the code I see the authors have addressed this with a subtle permutation in how groups are formed: https://github.com/637t0yvgcrmw/code01/blob/219f6e1ad4d38826401c9e37c13c6678222d26aa/modules.py#L214-L215 If I understand correctly, this permutation splits the overall channel dimension into two halves, then groups each half into groups of size n_split / 2. It then merges pairs from the two halves and permutes those, mixing across halves. This subtlety is highly non-obvious and should be mentioned in the main manuscript if space permits. The supplemental materials Figure 8c should also be modified to make this mixing across the two halves of the channel dimension clearer.

Clarity: The paper is well written and easy to comprehend.

Relation to Prior Work: The relationship to prior work is well explicated.

Reproducibility: Yes

Additional Feedback: Thank you to the authors for the clarifications and promised updates. I've adjusted my score from 6 to 7 since most of my concerns are now addressed.


Review 4

Summary and Contributions: This paper proposed a TTS model that converts input text to latent representation, which is then converted to mel spectrogram by a decoder. During the alignment, dynamic programming is used to compute the most probable alignment given the current model, then the alignment is used to update the model parameters. Besides, the invertible mel decoder enables the inference of latent representation from ground truth mel. This iterative method is very interesting. The trained model performs well on very long sentences.

Strengths: 1. The paper incorporates the monotonic property of text-to-speech in the model, which eliminates looping issues in many existing TTS models. 2. The model uses an iterative approach for training, which is novel in neural TTS. 3. The generated speech sounds good, and speech rate control is quite effective.

Weaknesses: The proposed model did not study emotion control. It would be good to study how emotion tokens (angry, excited) can be learned with some model extension.

Correctness: The problem formulation, model design, and evaluation look good to me.

Clarity: Yes.

Relation to Prior Work: The previous contributions were well covered. The training procedure of the proposed model is simpler compared to other parallel TTS models.

Reproducibility: Yes

Additional Feedback: When the flow-based decoder is introduced in the paper, it would be good to mention that latent code and mel have the same length. It will make the paper easier to read for people who are unfamiliar with flow-based models.

[Author Response · NeurIPS 2020]

Thanks all the reviewers for the detailed and thoughtful comments.

**(R1, R2) Resemblance to HMM-based speech synthesizers and recognizers** We found resemblance to the previous HMM-based works [1, 2, 3], all of which proposed methods to estimate alignments from unsegmented data. Thanks for commenting the missing references, we will mention about the previous works in the new version of the paper. However, we claim that the contribution of our proposed method is not only about the efficient alignment search, but also about enabling parallel sampling over the complex data distribution. We believe incorporating bijective flows for parallel sampling does not naturally come out from HMM-based works, as HMMs are inherently sequential models. Likewise, estimating alignments in the latent space through dynamic programming is distinctive from the previous parallel TTS works such as FastSpeech. Finally, compared to HMM-VAE [3], Glow-TTS differs in that 1) it focuses on generating samples in parallel not estimating sequences of latent discrete variables, 2) does not need an additional encoder to approximate the latent posterior nor 3) independence assumption of outputs across time.

**(R1, R3) Deterministic duration prediction** Our main goal was to build a parallel TTS model which jointly learns to model data distribution and align by itself in an efficient way, and the key components of Glow-TTS were the MAS and the flow-based decoder. We've not thoroughly explored to improve the duration predictor and simply follow the same architecture of FastSpeech. But as the feedback of Reviewer 1, exploring a joint model of durations would be promising in the future direction of our work and we expect it could be effective to model diverse speech.

**(R1, R3) Grouped 1x1 Convolutions** We design the grouped 1x1 convolutions to be able to mix channels. For each group, as Reviewer 3 commented, the same number of channels are extracted from one half of the feature map separated by coupling layers and the other half, respectively. For example, if a coupling layer divides a 8-channel feature map (a, b, c, d, e, f, g, h) into two halves (a, b, c, d) and (e, f, g, h), we implement to group them into (a, b, e, f) and (c, d, g, h) when the number of groups is 2. We will add the implementation details and update Figure 8c more clearer.

**(R1) Importance of parallel TTS** With parallel vocoders such as WaveGlow, the primary computational bottleneck in mel spectrogram-based end-to-end TTS systems became TTS models. For example, to generate a speech of 5.8 seconds on a GPU, WaveGlow spends 120 ms while Tacotron 2 and Glow-TTS require 640 ms and 40 ms, respectively. Therefore, adopting parallel TTS models significantly improves the sampling speed of end-to-end systems.

**(R3, R4) Controllability and diversity of sampling** We agree Glow-TTS shows restricted controllability and diversity of some properties of speech such as pitch or intonation as Reviewer 3 pointed out. But, as Glow-TTS estimates the distribution of speech given text tokens, it has the potential to generate diverse speech and control the characteristics of speech, which is not possible with the deterministic models like FastSpeech. The stochasticity comes from the prior distribution of the latent variable $z$, and we believe that synthesizing with careful sampling from the latent space enables us to control the characteristics of generated samples. In Section 5.3, we showed that varying temperature can change the pitch of generated samples and different $z$s correspond to speech samples with different intonations. To clarify our analysis, we will add more samples for a variety of text samples in our demo page. In this work, we provided the limited analysis on the controllability of Glow-TTS. Further analysis on how the latent space of $z$ is related to the characteristics of generated samples would be helpful for emotional control, and is left as future work.

**(R3) The iterative objective** The reason why we demonstrate the iterative objective provides a good lower bound of the global solution is that the iterative procedure is actually one example of widely used Viterbi training [2], which maximizes log likelihood of the most likely hidden alignment. We also empirically showed that our system works well with the objective. We will add a reference about Viterbi training.

**(R3) CMOS results** We conducted CMOS evaluation between Tacotron 2 and Glow-TTS with the sampling temperature 0.333. Through 500 ratings on 50 items, Glow-TTS wins Tacotron 2 by a gap of 0.934, which shows preference towards our model over Tacotron 2. We will add this result in the new version of the paper.

**(R2, R3) Multi-speaker MOS evaluations** We will add comparison to Tacotron 2 with speaker embeddings as a baseline in the new version of the paper. As for methodology of MOS evaluations, we randomly sampled 50 sentence and speaker id pairs from the test set, which resulted in 40 unique speakers (one sentence for 31 speakers, two sentences for 8 speakers, and three sentences for one speaker). When compared to Tacotron 2 with speaker embeddings, we will sample one sentence for each speaker for the evaluation. We will also include the test phrases as well as speaker ids used for all MOS evaluation in the supplemental material.

[1] Tokuda, Keiichi, et al. "Speech synthesis based on hidden Markov models." Proceedings of the IEEE 101.5 (2013): 1234-1252.

[2] Rabiner, Lawrence R. "A tutorial on hidden Markov models and selected applications in speech recognition." Proceedings of the IEEE 77.2 (1989): 257-286.

[3] Ebbers, Janek, et al. "Hidden Markov Model Variational Autoencoder for Acoustic Unit Discovery." INTERSPEECH. 2017.


[Meta-Review · NeurIPS 2020]

After rebuttal and discussion, all four reviewers provide very favorable reviews. The reviewers point out a novel methodology, combining flows with dynamic programming (hard monotonic alignment). The paper is therefore accepted for an oral.